# ICRA: A study of highly accurate course recommendation models incorporating false review filtering and ERNIE 3.0

Bing Li [1,2,3]*, Yuqi Hou[1,2,3], Jiangtao Dong[4], Biao Yang[5], Xile Wang[1,2,3]

**1** School of Software, Jiangxi Normal University, Nanchang, Jiangxi, China, **2** Research Centre for Management Science and Engineering, Jiangxi Normal University, Nanchang, Jiangxi, China, **3** Jiangxi Provincial Engineering Research Center of Blockchain Data Security and Governance, Nanchang, Jiangxi, China, **4** School of Computer and Information Engineering, Jiangxi Normal University, Nanchang, Jiangxi, China, **5** School of Digital Industries, Jiangxi Normal University, Nanchang, Jiangxi, China

* 004970@jxnu.edu.cn

**Data Availability Statement:** To comply with PLOS's data storage standards, we have uploaded the research dataset to the recommended public repository Figshare. The dataset's access link and

## Abstract

The rapid expansion of online education platforms has led to an influx of false reviews, complicating users' ability to identify suitable courses promptly. Addressing these challenges, this paper introduces ICRA (Intelligent Course Review Analysis), a novel model that identifies and filters false reviews using a custom sentiment lexicon and a pre-trained ERNIE 3.0 model. ICRA enhances data quality by analyzing user reviews and course profiles comprehensively for recommendation purposes. The model utilizes the BERT lexicon and ERNIE 3.0 to obtain deep semantic representations. It integrates BiLSTM with a multi-head attention mechanism to capture essential review details, aiming to minimize overfitting and enhance generalization. By predicting user review scores and verifying review authenticity, ICRA boosts recommendation accuracy and robustness, addressing the cold-start issue. Experimental findings highlight ICRA's excellence in predicting user ratings and delivering precise course recommendations efficiently. This capability streamlines course selection on online education platforms, improving learning experiences and efficiency.

## Introduction

Online education has undergone significant innovation amidst global digitization and intelligence trends. It meets the demand for fragmented learning and promotes educational equity by overcoming geographical limitations, laying a foundation for a lifelong learning society. Concurrently, the proliferation of MOOC (Massive Open Online Courses) platforms like Coursera, edX, and Udacity [1] has presented learners with numerous course options, posing both opportunities and challenges. The surplus of course offerings has led many learners to experience "course overload" and "course disorientation", shifting them from having "no course to choose" to feeling they have "no way to learn". Learners consider several key factors when selecting a course, such as course content quality, instructor credentials, and user feedback [2]. As shown in Fig 1, after completing the course, users provide feedback, which helps other users make more informed choices.

DOI are as follows: https://doi.org/10.6084/m9.figshare.27301371.v1.

**Funding:** The authors acknowledge the financial support provided by the National Natural 446 Science Foundation of China (72161020), the Jiangxi Provincial Natural Science 447 Foundation (20224BAB202023), the Jiangxi Social Science Foundation Project 448 (21GL44), the Science and Technology Research Project of Jiangxi Provincial Education 449 Department (GJJ2200333).

Personalized recommendation systems have emerged, intelligently generating course recommendations tailored to learners' needs and interests by analyzing their historical data to optimize learning paths and enhance learning efficiency [3]. Currently, recommendation technologies in MOOCs primarily focus on content-based, knowledge-based, and hybrid recommendations [4]. Each approach has unique characteristics and corresponding challenges. Content-based recommendation methods recommend courses based on course content features and learner preferences [5]. Researchers have proposed innovative methods, such as employing graph convolutional neural networks to handle implicit "user-course" interactions [6], integrating user attributes from MOOC platforms [7], and introducing Bayesian personalized ranking [8] to address challenges like complex feature extraction and data sparsity [9]. Knowledge-based recommendation methods utilize domain knowledge and expert intelligence to construct logical learning paths, particularly beneficial for mitigating the cold-start problem using methods like evolutionary search algorithms [10], collaborative filtering with FP-growth algorithms [11], multidimensional knowledge graph frameworks [12], and knowledge graph embedding techniques [13]. Hybrid recommendation strategies integrate multiple methods to offer more personalized and comprehensive recommendation services, despite increasing system complexity. Researchers use machine learning and association rule mining for interest matching [14], develop Moodle plugins [15], and apply fuzzy logic and the xDeepFM model to continuously recommend relevant courses [16], effectively addressing the cold-start problem and providing precise recommendations for users.

Despite achieving remarkable results, existing online education recommendation systems have gradually revealed limitations due to the increasing diversity of user needs and complexity of course content. User comments directly provide feedback on course quality, teaching content, and teachers' methods, containing valuable information crucial for evaluating course quality and teaching effectiveness. However, most current MOOCs recommender systems underutilize user comments and fail to fully explore these valuable resources [17]. On MOOC platforms, some users may post false comments, misleading others and compromising the accuracy of recommender systems. Research on course recommendation systems generally focuses on two key perspectives: user reviews and course reviews. The user review perspective

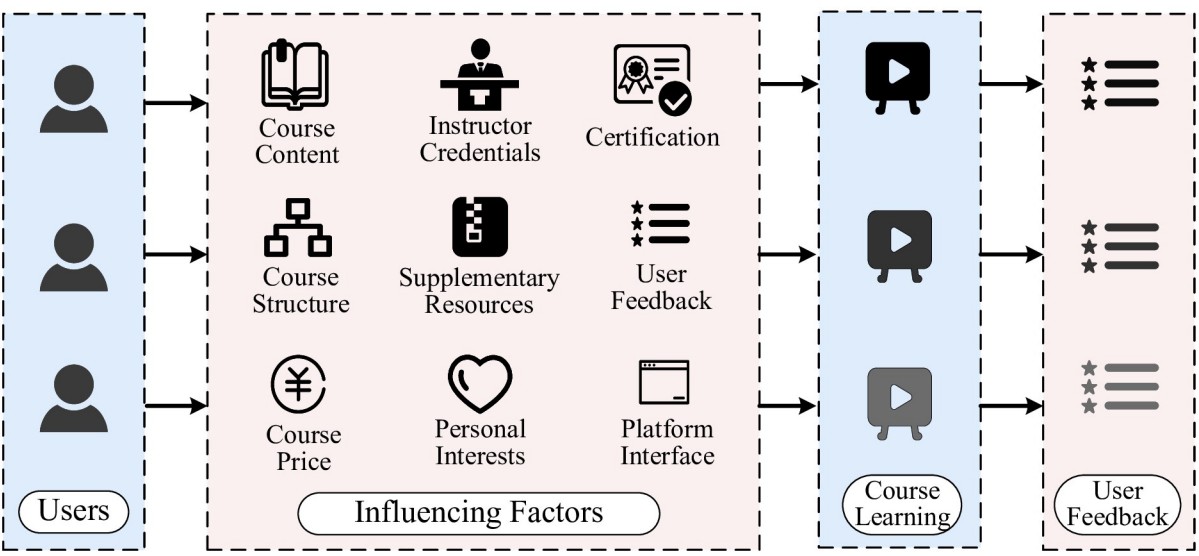

**Fig 1. Course selection decision flowchart.**

analyzes users' evaluations and feedback, extracting valuable insights such as review content and emotional tone to better understand user preferences and needs, thereby constructing a more accurate user profile. On the other hand, the course review perspective assesses the quality and effectiveness of the course, analyzing factors like review content and course ratings to provide a more precise course profile. Some studies have shown innovative approaches and results. Some studies have demonstrated innovative methods and achievements. For instance, researchers have introduced counterfactual sample enhancement techniques to improve the performance of these systems [18]. Additionally, BERT and Bi-GRU networks have been combined to learn semantic features and sentiment weights from review texts [19]. Moreover, a novel Review-based Graph Comparison Learning (RGCL) framework has been proposed to further enhance recommendation accuracy [20].

However, varying review quality, lack of professionalism and depth, false reviews, and weak user-item review correlation continue to hinder the effectiveness of review information. Hence, future research should concentrate on effectively integrating user and item information using natural language processing, sentiment analysis, and other technical means to enhance recommender system accuracy, reliability, and personalized recommendation services. Fig 2 compares traditional recommender systems with the ICRA recommendation model. Traditional recommender systems primarily rely on user reviews and ratings, but inconsistencies between reviews and ratings often affect system reliability. To address this, the ICRA model integrates false review filtering with course overview analysis, improving recommendation accuracy and reliability.

The primary contributions of this study include:

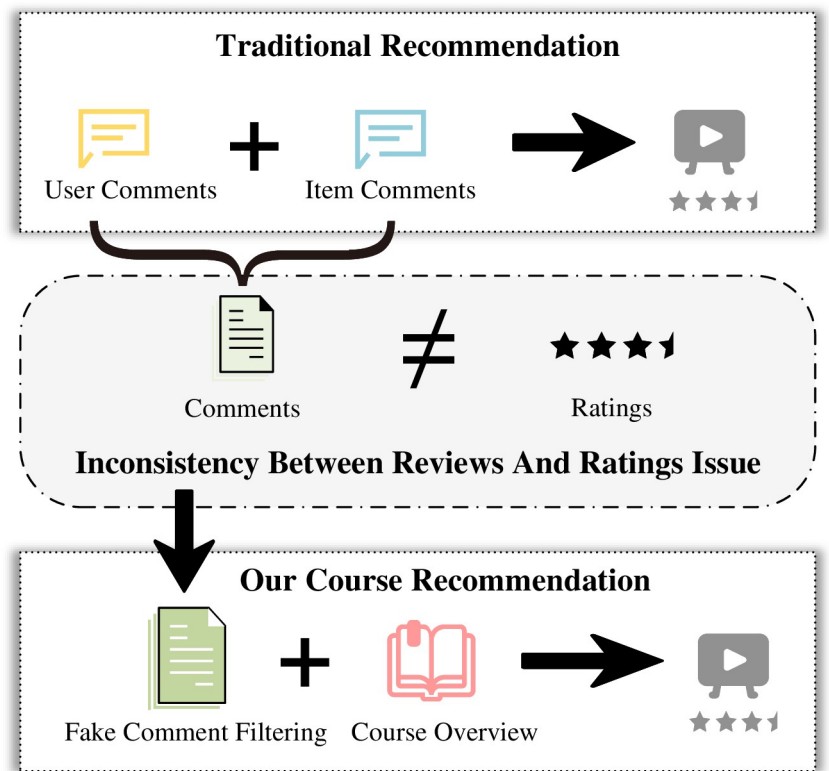

**Fig 2. Problems with traditional course recommendations.**

1. Data from Chinese MOOC universities was collected via web crawlers, and the identification and filtering of false reviews improved data quality, enriching the dataset.

2. The ICRA model integrates review and course information, performing semantic analysis to more accurately understand user needs and preferences.

3. Multiple experiments validated the effectiveness of the ICRA model, significantly improving the accuracy and robustness of the recommendation system.

## Related work

### Pre-training model ERNIE3.0

From early statistical models to advanced neural networks like BERT and GPT, pre-trained language models have greatly improved data efficiency, generalization, and enabled transfer learning. While challenges remain in common-sense and logical reasoning, Baidu's ERNIE series (Enhanced Representation through Knowledge Integration) advances language understanding by integrating knowledge graphs and large-scale corpora. ERNIE 1.0 outperforms BERT in Chinese tasks [21], and ERNIE 2.0 excels in both English and Chinese tasks through multi-task learning [22]. ERNIE 3.0, which integrates knowledge graphs, achieved outstanding results in the 2021 SuperGLUE benchmark [23].

As a pre-trained model, ERNIE 3.0 processes tokenized and segmented text sequences by combining word embeddings, position embeddings, and segment embeddings to generate context-aware semantic representations. First, the preprocessed text is fed into the Embedding layer to generate the word embedding vectors $W = \{w_1, w_2, \ldots, w_n\}$. Then, the encoding layer extracts features, resulting in an output vector matrix $X = \{x_1, x_2, \ldots, x_n\}$. This matrix can be used as input for downstream tasks such as text generation, named entity recognition, and text classification. The architecture of the ERNIE 3.0 model is illustrated in Fig 3.

### Attention mechanism

The attention mechanism dynamically directs focus to crucial information at different positions within the input sequence, thereby optimizing the efficiency and accuracy of model processing. Central to the attention mechanism is the mapping of each element in the input sequence $X$ to Query, Key, and Value vectors using trainable weight matrices $W_Q$, $W_K$, and $W_V$ [24].

$$Q = XW_Q, \quad K = XW_K, \quad V = XW_V \tag{1}$$

The similarity between the query vector $Q$ and the key vector $K$ is calculated by scaling the dot product, followed by applying the softmax function to normalize attention weights. These weights are then used to weight the summed value vectors, thereby extracting pivotal information from the input sequence:

$$\text{Attention}(Q, K, V) = \text{softmax}\left(\frac{QK^T}{\sqrt{d_k}}\right) V \tag{2}$$

Here, $d_k$ denotes the dimensionality of the key vector, and it scales the dot product to mitigate gradient vanishing. The softmax function ensures that the attention weights sum up to one.

The multi-head attention mechanism, a variant of the attention mechanism, significantly enhances the model's ability to capture intricate patterns and diverse inputs. By computing

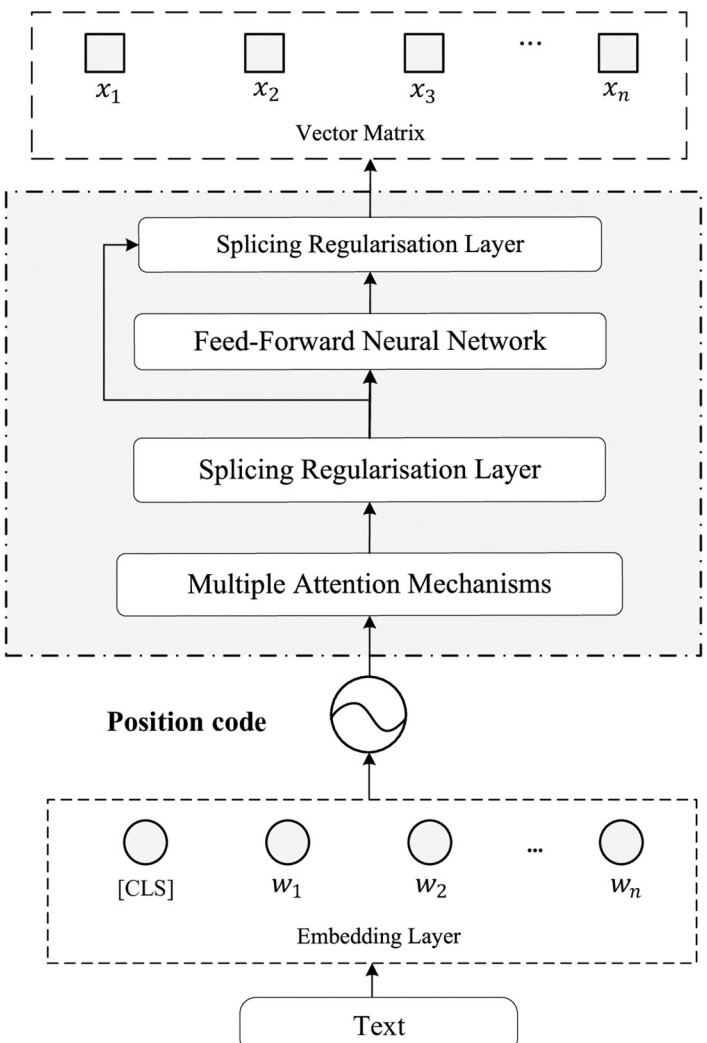

**Fig 3. ERNIE 3.0 model framework diagram.**

multiple independent attention heads in parallel, each focusing on distinct features of the input sequence, it captures richer and more comprehensive contextual information. Each attention head independently computes attention weights and weighted values:

$$\text{head}_i = \text{Attention}(QW_{Q_i}, KW_{K_i}, VW_{V_i}) = \text{softmax}\left(\frac{(QW_{Q_i})(KW_{K_i})^T}{\sqrt{d_k}}\right)VW_{V_i} \quad (3)$$

The outputs of all attention heads are concatenated and undergo linear transformation to yield the final representation of multi-head attention:

$$\text{MultiHead}(Q, K, V) = \text{Concat}(\text{head}_1, \text{head}_2, \ldots, \text{head}_h)W_o \quad (4)$$

Here, $h$ is the number of attention heads, and $W_o$ is the concatenated linear transformation matrix. The multi-head attention mechanism empowers the model to capture multiple relevant features within the input sequence, thereby enhancing its capacity to address complex tasks.

## A study of fake reviews

Research indicates that many online product review platforms are plagued by fake reviews. Approximately 16% of reviews on Yelp and 42% on Amazon are estimated to be fake [25]. According to the "China Internet Users' Rights Protection Report", 72.7% of consumers have encountered fake reviews [26]. Fake reviews not only mislead consumer decisions and disrupt fair market competition, but also undermine platform credibility and lead to significant economic losses. The "2021 Fake Online Review Report," jointly released by CHEQ and University of Baltimore Professor Roberto Cavazos [27], estimated that fake reviews affected $152 billion in consumer spending in 2021. Academic research primarily focuses on the motivations, identification methods, and impacts of fake reviews. Online Deceptive Reviews (ODRs) are harmful to the e-commerce ecosystem due to factors such as emotional venting, incentive-driven reviews, perfunctory comments, and "water armies" manipulating merchant reputations [28]. To tackle fake reviews, researchers have proposed innovative approaches, such as n-gram and sentiment scoring-based intelligent systems [29], sentiment analysis integrating habit bias and XGBoost algorithms [30], and methods like the ERNIE pre-trained model and convolutional neural networks [31].

As a part of online products, online education is also threatened by fake reviews. Recommender systems rely on user ratings and other metrics, and fake reviews can significantly undermine the accuracy and reliability of recommendation results. Competitors may post fake reviews to discredit rival courses or artificially inflate their own ratings, misleading learners in their decision-making. Institutions or individuals might also employ "water armies" to post fake positive reviews to attract more learners. These actions not only harm learners' interests but also weaken the credibility of the recommendation system. Applying false review identification technology to course recommendation systems is essential for ensuring accuracy, protecting learners' interests, and fostering a fair and reliable online education environment.

## Research methodology

### Data collection

Current MOOC platform research lacks publicly available, comprehensive, and standardized datasets. Existing datasets are often constrained by data restrictions, limited scope, and inconsistent formats across platforms, which hinders in-depth analysis of user behavior patterns and learning trends. To address this gap, we employed Selenium and BeautifulSoup tools to scrape data from the China University MOOC platform, which offers over 10,000 open courses, including more than 1,400 nationally recognized quality courses across various disciplines. By simulating user clicking and page-turning behaviors, we systematically collected key information such as user IDs, comment texts, ratings, timestamps, course IDs, and detailed course descriptions. This dataset comprises 4,347 courses, 534,291 users, and 747,253 comments. To ensure privacy protection, all data were anonymized and de-identified to prevent any disclosure of personal information. The dataset is divided into raw and preprocessed versions, with the latter undergoing invalid data removal, text normalization, and sentiment labeling for training in the course recommendation model. This study strictly adhered to the platform's terms of service and privacy policy, collecting only publicly available, non-sensitive information solely for academic research purposes.

### Data preprocessing

A systematic cleaning and preprocessing of user data from Chinese university MOOC platforms was conducted to ensure data quality and reliability, thereby laying a solid foundation

**Table 1. Partial course information.**

| Course ID | Course Overview |
|---|---|
| 1 | This is an introductory course on computer algorithms. |
| 2 | This course is aimed at doctoral students and young researchers with a background in cell biology and developmental biology. |
| 3 | Dental Anatomy and Oral Physiology is an important foundational course in dental medicine. |

for subsequent analyses. Based on the cleaning methods outlined below, a total of 194,352 comments were cleaned, retaining 552,901 valid data entries. The processed data is presented in Tables 1 and 2.

**Missing values processing**. Records lacking learning hours and user names are deleted directly. For courses marked as "No rating," the average rating value of all course comments is used to fill in the rating field.

**Format standardization**. Convert all date and time fields to ISO 8601 standard format for data readability and comparability. Ratings were standardized to a 1 to 5 scale.

**Filtering useful data**. Remove records where user activity data (e.g., learning hours, number of followers, number of fans, etc.) are all zero. Retain only the record with the longest learning hours for each user name to reflect their actual learning status.

**Handling outliers**. Extreme learning hours records are treated as data entry errors or technical issues; a reasonable maximum threshold (not exceeding 10 years) is set for learning hours, and unrealistic records are adjusted or deleted.

**Text pre-processing**. Reduce the weight of comment text, eliminate mechanical filler words and unnecessary phrases, enhance text data quality, and prepare clean input for natural language processing and sentiment analysis.

## Build your own emotional dictionary

To enhance sentiment analysis accuracy and applicability for MOOC user comments, this study employs the SO-PMI algorithm to construct a domain-specific sentiment lexicon in education. Current sentiment lexicons lack coverage of domain-specific sentiment expressions. The self-built lexicon accurately assesses sentiment polarity of candidate words in education using statistical and lexical co-occurrence data, better reflecting MOOC user sentiment responses.

During data processing, the jieba.posseg tool annotated and segmented lexically to filter out words of practical significance. Text was extracted from randomly selected MOOC and Tencent Classroom review datasets, emotional keywords were screened based on word frequency, and words with insignificant emotional tendencies were manually eliminated.

Remaining words are categorized into positive or negative sentiment thesauri, with known sentiment words added to the jieba thesaurus to enhance word separation accuracy. A fixed window size (5 words) was used for context scanning around known sentiment words, collecting co-occurring words and constructing a word pair co-occurrence matrix for PMI value

**Table 2. Partial user information.**

| User ID | Timestamp | Comment | Course ID | Rating |
|---|---|---|---|---|
| 441528 | 2023-05-05 | Gained more understanding of human and regenerative medicine | 2 | 5 |
| 448029 | 2023-04-14 | Learned very practical theoretical knowledge | 2 | 5 |
| 494744 | 2023-04-05 | Strong professional knowledge, good teaching ability, clear and understandable | 2 | 4 |

calculation. Pointwise Mutual Information (PMI) between $w_1$ and $w_2$ is computed as:

$$\text{PMI}(w_1, w_2) = \log_2\left(\frac{P(w_1 \cap w_2)}{P(w_1) \times P(w_2)}\right) \qquad (5)$$

Where $P(w_1 \cap w_2)$ denotes the probability of co-occurrence of words $P(w_1)$ and $P(w_2)$ represent their respective independent probabilities.

By comparing the PMI values of word pairs with positive and negative sentiment words, the Semantic Orientation Pointwise Mutual Information (SO-PMI) value for each candidate word is calculated as:

$$\text{SO-PMI} = \sum_{\text{pos}} \text{PMI}(\text{pos}, \text{candi\_word}) - \sum_{\text{neg}} \text{PMI}(\text{neg}, \text{candi\_word}) \qquad (6)$$

The sentiment polarity of candidate words is determined based on computed results: a positive SO-PMI value indicates a positive word, and a negative value indicates a negative word.

## Fake comment identification

Through the analysis of the dataset, we identified several anomalies, as shown in the Table 3: some positive reviews were accompanied by low ratings, while, conversely, some negative reviews were given high ratings. There were also cases where reviews consisted solely of punctuation marks or meaningless characters. Additionally, we observed that certain users repeatedly used similar wording to review the same course. To address these issues, the fake review detection module consists of three main components, employing a multidimensional approach to effectively identify fake reviews:

- **Inconsistency between ratings and review content**: First, the module uses sentiment analysis to compare the course rating with the review content, generating a sentiment-based rating. If there is a significant discrepancy between this rating and the actual rating, the review is flagged as a potential fake.

**Table 3. Partial anomalous comments.**

| User ID | Comment | Course ID | Rating |
|---------|---------|-----------|--------|
| 521955 | Teacher Zhou explains very well | 151 | 1 |
| 444345 | Explains very well | 152 | 1 |
| 436343 | Explains in great detail, very beneficial | 192 | 1 |
| 458956 | The course is rich, with abundant resources | 159 | 1 |
| 263667 | It's trash! | 433 | 5 |
| 73009 | Trash wanyi | 455 | 5 |
| 212322 | Trash trash trash | 4147 | 5 |
| 4 | Gained knowledge, studied | 2 | 5 |
| 4 | Gained knowledge, studied | 19 | 5 |
| 4 | Gained knowledge, studied. | 956 | 4 |
| 4 | Gain knowledge, study. | 707 | 5 |
| 4 | Gained knowledge, studied. | 2398 | 4 |
| 431802 | ;,;l,>m | 94 | 5 |
| 423644 | ?.?.?. | 100 | 5 |
| 433809 | ,.kjlhhio | 100 | 4 |

- **Meaningless or excessively short reviews**: Next, the module filters reviews based on their length and the use of symbols. Reviews that are too short [29], contain abnormal symbols, or include garbled text are considered meaningless or superficial and are thus removed.

- **Highly templated or duplicate reviews**: Lastly, the module applies text similarity analysis to identify highly similar or duplicate reviews. The ERNIE 3.0 model is used to perform deep semantic encoding of the review text, which is then tokenized using the BERT tokenizer for further processing. The Annoy library is employed to build an approximate nearest neighbor index, enabling efficient similarity comparisons to detect duplicate reviews.

By combining review length, sentiment analysis results, and similarity analysis, the module can accurately detect and filter fake reviews. All evaluation results are recorded and stored for further screening and analysis. The module structure is shown in Fig 4.

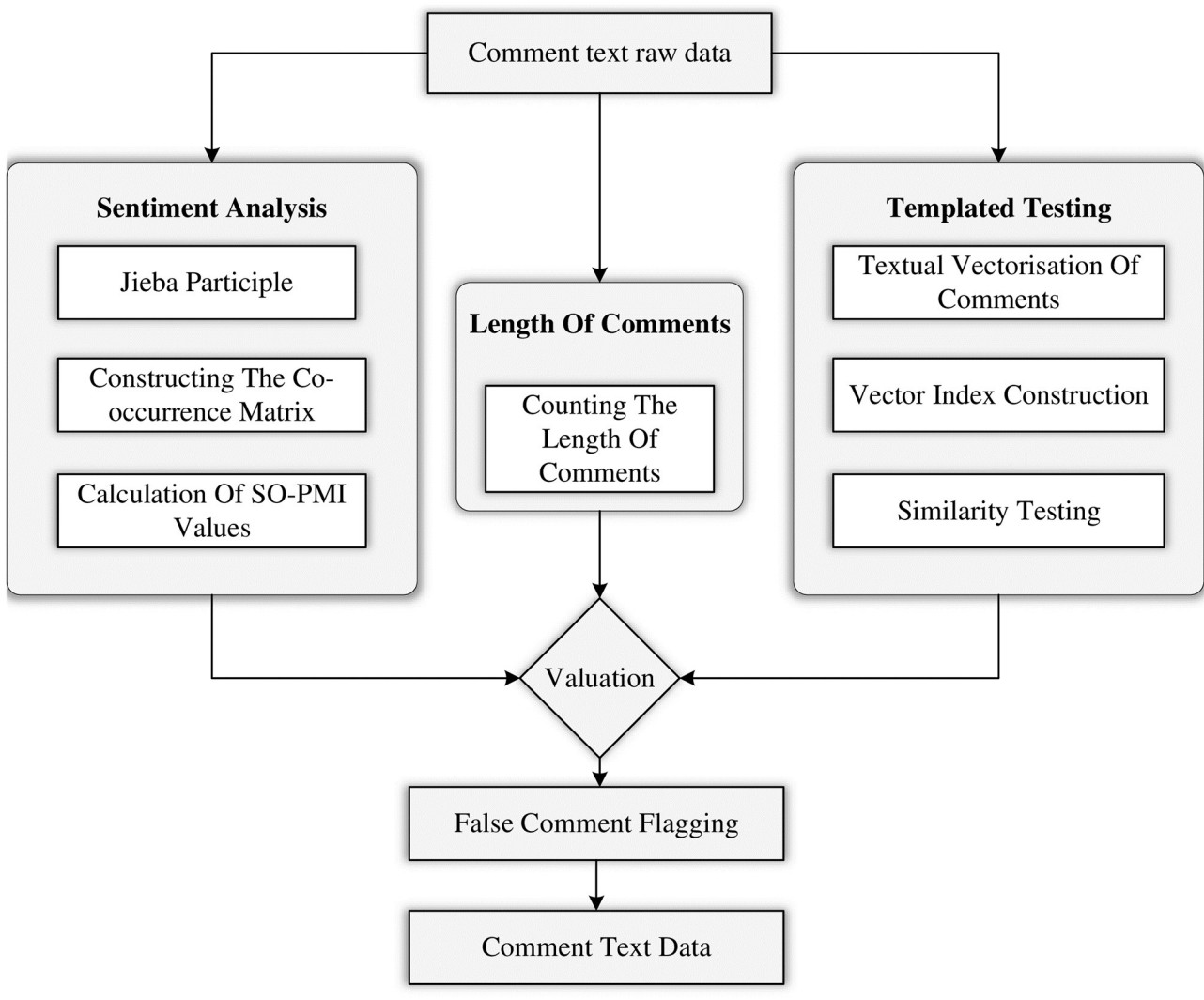

**Fig 4. False comment recognition module.**

## Model architecture

Rating prediction is one of the core problems in personalized recommendation systems, aiming to score the interactions between users and courses. The ICRA model predicts user ratings for unseen courses, providing a basis for personalized ranking in the recommendation system. The model predicts scores for all candidate courses and ranks them based on the predicted scores, prioritizing those with higher ratings to achieve personalized recommendations. The ICRA model leverages data collected from the Chinese University MOOC platform, combined with a self-built sentiment lexicon and the pre-trained ERNIE 3.0 model, to conduct in-depth semantic analysis. This model not only accurately identifies and filters fake reviews, ensuring high-quality datasets, but also predicts user ratings for new courses based on their historical rating records. To achieve this, the ICRA model consists of an embedding layer, a feature processing layer, a regularization layer, and an output layer, all of which work closely together to improve rating prediction accuracy.

Fig 5 illustrates the overall architecture of the ICRA model, where $C_u$ represents the user comment document, $C_i$ represents the course overview document, $E_u$ and $E_i$ are the embedding matrices for user and course comment documents, respectively, and $y_i$ and $\hat{y}_i$ denote the actual and predicted ratings of users for courses.

**Embedded layer.** The embedding layer's primary function is to convert the comment text $C_u$ of user $U_p$ and the overview text $C_i$ of course $I_q$ into vector representations suitable for model processing. Here, $P$ and $Q$ denote the number of users and courses, respectively. Initially, the BERT tokenizer tokenizes $C_u$ and $C_i$, and batch processing is performed on GPU for efficiency. The BERT tokenizer handles various language phenomena and generates context-aware word vectors. These tokenized outputs are then fed into the ERNIE 3.0 model to obtain vector representations for each word. Subsequently, the embedding vector of each comment is computed by averaging all word vectors within that comment. Specifically, the embedding vector $e_{u_p}^{(l)}$ of the $l$-th comment of user $U_p$ is computed as the mean of all word vectors $v_1, v_2, \ldots, v_n$, where $n$ denotes the number of words in the comment:

$$e_{u_p}^{(l)} = \frac{1}{n} \sum_{k=1}^{n} v_k \tag{7}$$

Here, $v_k$ is the vector representation of the $k$-th word, and $v_k \in \mathbb{R}^w$, where $w$ is a hyperparameter representing the dimensionality of the text vectors. Combining all embedding vectors of user $U_p$'s comments forms a matrix, resulting in the embedding matrix $E_u$, where each row represents the embedding vector of a comment. Similarly, for the course $I_q$'s overview text, an embedding matrix $E_i$ is obtained. Through these steps, embedding matrices for user comment documents and course overview documents are obtained, which serve as inputs for subsequent model processing.

**Feature processing layers.** The feature processing layer further processes the vector representations generated by the embedding layer to capture important features and dependencies in the text.

*Bi-LSTM Layer.* The sequence features from the ERNIE 3.0 model are processed through a bidirectional LSTM (`nn.LSTM`) to capture both forward and backward contextual dependencies within the text. The bidirectional LSTM consists of two components:

$$(\mathbf{h}_t)^{\rightarrow} = \text{LSTM}^{\rightarrow}(E_{u,t}, (\mathbf{h}_{t-1})^{\rightarrow}) \tag{8}$$

$$(\mathbf{h}_t)^{\leftarrow} = \text{LSTM}^{\leftarrow}(E_{u,t}, (\mathbf{h}_{t+1})^{\leftarrow}) \tag{9}$$

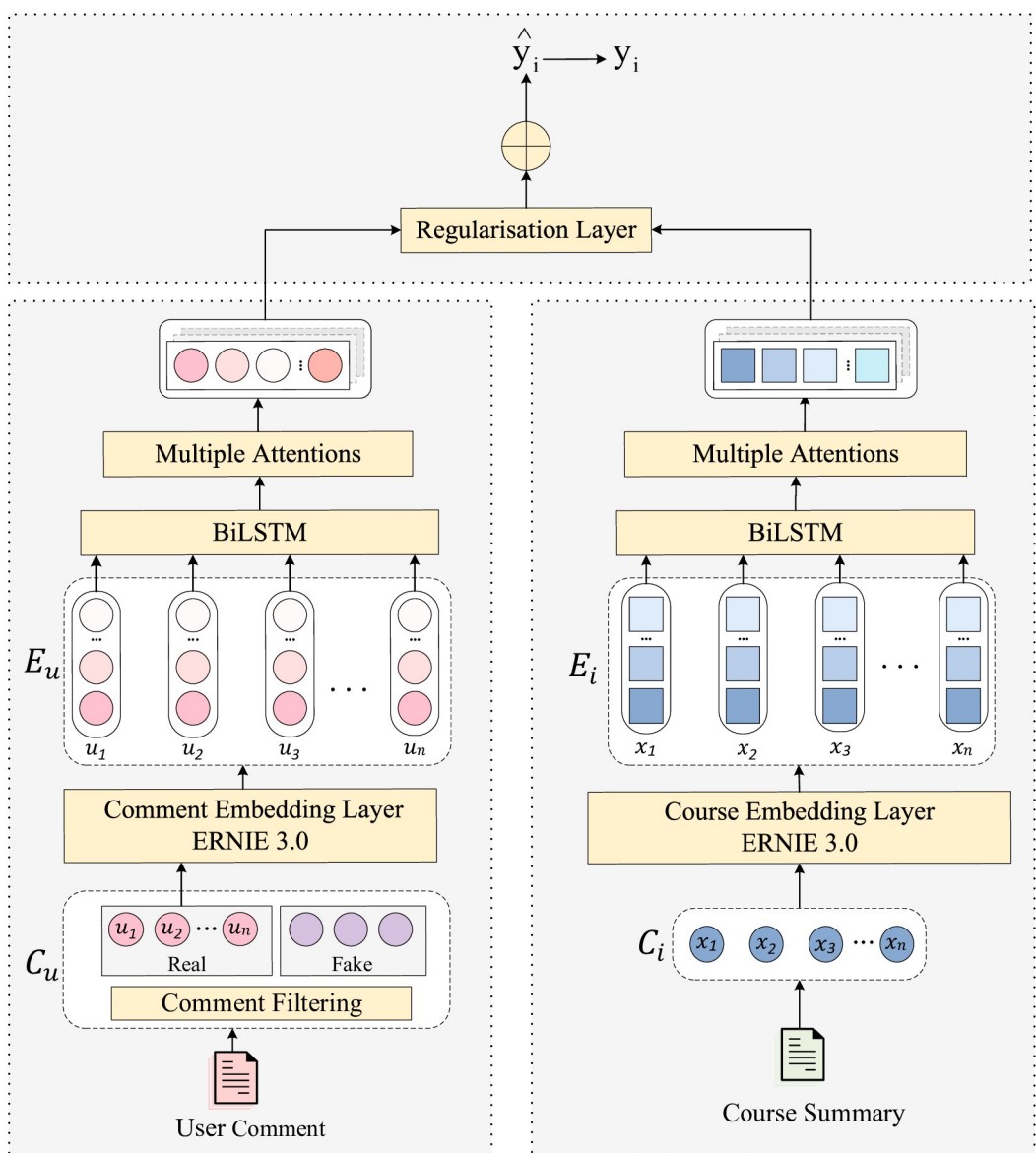

**Fig 5. Model architecture diagram.**

where $E_{u,t}$ denotes the embedding vector of the user at time step $t$, and $(\mathbf{h}_t)^{\rightarrow}$ and $(\mathbf{h}_t)^{\leftarrow}$ represent the hidden states of the forward and backward LSTMs at time step $t$, respectively. The final output of the bidirectional LSTM is:

$$\mathbf{H}_{u,\text{lstm}} = [\mathbf{H}_{u,\text{lstm},1}, \mathbf{H}_{u,\text{lstm},2}, \dots, \mathbf{H}_{u,\text{lstm},T}] \tag{10}$$

where $\mathbf{H}_{u,\text{lstm}}$ denotes the concatenated bidirectional LSTM outputs over the sequence length $T$. Similarly, $\mathbf{H}_{i,\text{lstm}}$ represents the bidirectional LSTM output for the course.

*Multi-Head Attention Mechanism.* This mechanism enhances context-awareness by focusing on significant information within the LSTM outputs. Detailed explanations are provided in Section 2.2 and are omitted here for brevity. $A_u$ and $A_i$ represent the attention weights for

user and course comments.

$$\mathbf{A}_u = \text{MultiHeadAttention}(\mathbf{H}_{u,\text{lstm}}) \tag{11}$$

$$\mathbf{A}_i = \text{MultiHeadAttention}(\mathbf{H}_{i,\text{lstm}}) \tag{12}$$

**Regularisation layer.** The regularization layer includes pooling, normalization, and dropout layers, aimed at reducing overfitting, enhancing generalization, and maintaining stability during the training process.

*Pooling layer.* The attention outputs are pooled using global average pooling (`nn.AdaptiveAvgPool1d`) to extract global features, reduce feature dimensions, and retain important information. For each feature vector $a_u \in \mathbf{A}_u$:

$$P_u = \frac{1}{T} \sum_{t=1}^{T} a_{u,t} \tag{13}$$

Here, $a_{u,t}$ represents the attention output feature vector of the user and course at time step $t$, $T$ is the length of the feature sequence, and $P_u$ is the pooled feature vector.

*Normalization layer.* Layer normalization (`nn.LayerNorm`) is applied to the pooled features to ensure training stability, preventing gradient vanishing or exploding issues. For each user feature vector $P_u$:

$$n_u = \frac{P_u - \mu_u}{\sigma_u + \epsilon} \cdot \gamma_u + \beta_u \tag{14}$$

Here, $\mu_u$ and $\sigma_u$ are the mean and standard deviation of the user feature vector, $\gamma_u$ and $\beta_u$ are the scaling and shifting parameters of the user feature vector, and $\epsilon$ is a small constant to prevent division by zero.

*Dropout layer.* The dropout layer (`nn.Dropout`) randomly drops some neurons to reduce interdependence among neurons, prevent overfitting, and enhance model generalization. For each feature vector $n_u \in \mathbf{N}_u$:

$$d_u = \text{Dropout}(n_u) \tag{15}$$

The dropout operation randomly sets some elements to zero while scaling the remaining elements to maintain the expected value of the overall feature vector unchanged:

$$d_{u,j} = \begin{cases} 0, & \text{if } m_{u,j} = 0 \\ \frac{n_{u,j}}{1-p}, & \text{if } m_{u,j} = 1 \end{cases} \tag{16}$$

Here, $m_{u,j}$ is the mask vector, and $p$ is the dropout probability.

**Output layer.** *Output layer.* The output layer of the model employs a Multi-Layer Perceptron (MLP) for non-linear transformation, generating the predicted rating. The MLP includes multiple neural network layers such as linear layers, ReLU activation functions, and Dropout layers. The model's final output, which predicts user ratings for courses, is computed as

follows:

$$F = [D_u; D_i] \tag{17}$$

$$F^{(1)} = \text{ReLU}(W_1 F + b_1) \tag{18}$$

$$F^{(2)} = \text{Dropout}(F^{(1)}) \tag{19}$$

$$\hat{y}_i = W_2 F^{(2)} + b_2 \tag{20}$$

Here, $F$ represents the fused feature vector, $W_1$ and $W_2$ denote the weight matrices of the fully connected layers, $b_1$ and $b_2$ represent the bias terms, and $\hat{y}_i$ signifies the final predicted rating.

## Experimental design and result analysis

### Experimental setup

The experimental data were divided into original and real datasets, with the original dataset containing 552,901 comments and the real dataset consisting of 490013 comments after filtering out fake reviews. These datasets were primarily used for rating prediction tasks. To evaluate the model's performance, RMSE (Root Mean Squared Error) and MAE (Mean Absolute Error) were used as the main evaluation metrics. Lower RMSE and MAE values indicate higher prediction accuracy, providing a comprehensive assessment of the algorithm's performance. The formulas are defined as follows:

$$\text{MAE} = \frac{1}{N} \sum_{i=1}^{N} |y_i - \hat{y}_i| \tag{21}$$

$$\text{RMSE} = \sqrt{\frac{1}{N} \sum_{i=1}^{N} (y_i - \hat{y}_i)^2} \tag{22}$$

Here, $N$ represents the number of samples, $y_i$ denotes the actual rating for sample $i$, and $\hat{y}_i$ signifies the predicted rating for sample $i$. These evaluation metrics provide a systematic and comprehensive assessment of the recommendation algorithm's accuracy and performance.

### Comparison experiment

To validate the effectiveness of the proposed algorithm, we compared it with seven different models. We conducted five experiments across all datasets and averaged their performance to assess the accuracy of each model on the datasets.

- **BERT** [32]: BERT (Bidirectional Encoder Representations from Transformers) is a bidirectional encoder model trained on extensive unsupervised text data. It generates context-aware word vectors, significantly enhancing the accuracy and robustness of recommendation systems.

- **ROBERTa** [33]: ROBERTa (Robustly Optimized BERT Approach) is an enhanced version of BERT, improving semantic understanding and recommendation system performance through extensive pre-training and hyperparameter optimization on a larger dataset.

- **DeepCoNN** [34]: DeepCoNN employs parallel convolutional neural networks to extract features from user and course reviews. These features are subsequently integrated through shared hidden layers to predict user ratings.

- **MPCN** [35]: MPCN (Multi-Pointer Co-Attention Networks) utilizes a multi-pointer co-attention mechanism to enhance recommendation accuracy by extracting crucial information from reviews and course descriptions. This approach ensures personalized recommendations through dynamic inference of review importance and multi-level interactions.

- **CARL** [36]: CARL (Context-Aware Rating Learning) employs a neural network framework with an attention mechanism to enhance the accuracy of rating predictions by learning context-aware user-product representations.

- **ATN** [37]: ATN (Attention-based Time-aware Network) utilizes LSTM for extracting review features, integrates CNN and self-attention mechanisms for learning user and product features, and incorporates temporal factors to enhance recommendation accuracy.

- **PREEI** [19]: PREEI is a personalized recommendation algorithm that incorporates sentiment expressiveness and review importance. It employs BERT for text vectorization, Bi-GRU for semantic feature learning, and integrates sentiment weights and attention mechanisms for rating prediction using the DeepFM algorithm.

Table 4 compares the RMSE and MAE of each model on the original and real datasets, demonstrating performance improvements. The "Difference" column represents the discrepancy between the original and real datasets, reflecting the performance changes after filtering out false reviews. The last row, "Improved," shows the percentage improvement of the ICRA model compared to the historically best methods. On the original dataset, ICRA's RMSE improved by 4.31% over ANT, while MAE improved by 2.54%. On the real dataset, RMSE improved by 6.73% over ANT, and MAE improved by 2.08% over BERT. The heatmap in Fig 6 provides a visual representation of model performance, where darker colors indicate better performance. Most models showed improvements on the real dataset, with the ICRA model showing the most significant RMSE and MAE improvements, as highlighted by the dark blue regions. These results demonstrate that the ICRA model achieved significant progress in predictive accuracy, particularly in addressing false reviews.

Compared to ICRA, the improvements in other models are relatively modest. DeepCoNN and MPCN show notable performance gains on real data after filtering out false comments, whereas PREEI exhibits minimal improvement, suggesting limitations in adaptability. The

**Table 4. Comparison experiment results.**

| Model | RMSE | | | MAE | | |
|---|---|---|---|---|---|---|
| | Original | Real | Difference | Original | Real | Difference |
| BERT | 0.515 | 0.362 | 0.15 | 0.359 | **0.289** | 0.07 |
| ROBERTa | 0.557 | 0.383 | 0.17 | 0.371 | 0.341 | 0.03 |
| DeepCoNN | 0.585 | 0.405 | 0.18 | 0.484 | 0.385 | 0.10 |
| MPCN | 0.564 | 0.416 | 0.15 | 0.484 | 0.371 | 0.11 |
| CARL | 0.514 | 0.384 | 0.13 | 0.417 | 0.379 | 0.04 |
| ANT | **0.441** | **0.312** | 0.13 | **0.355** | 0.297 | 0.06 |
| PREEI | 0.659 | 0.637 | 0.02 | 0.372 | 0.359 | 0.01 |
| ICRA (Ours) | **0.422** | **0.291** | 0.13 | **0.346** | **0.283** | 0.06 |
| Improved | 4.31% | 6.73% | - | 2.54% | 2.08% | - |

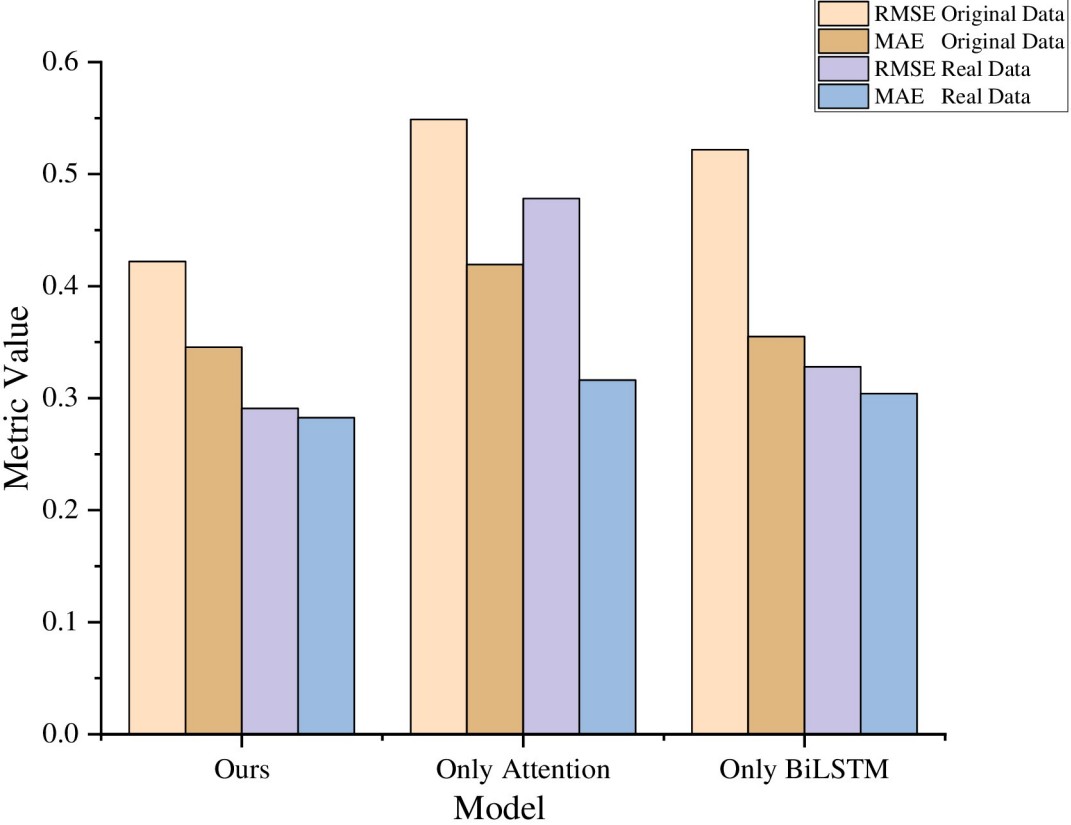

**Fig 6. Heatmap of model evaluation metrics.**

strong performance of the ANT model underscores the effectiveness of its attention mechanism in managing complex recommendation tasks.

The ICRA model's improvements in RMSE and MAE surpass those of most other models, demonstrating its robustness and accuracy in course recommendation tasks. This indicates that ICRA effectively captures deep user-course interactions, delivering precise recommendations even in challenging, noisy datasets.

## Ablation experiment

In the ablation study, we evaluated the contribution of various components to the overall model performance by removing or replacing key elements. Specifically, we designed two variant models:

- **Ours-BiLSTM**: In this variant, the BiLSTM layer was removed, leaving the multi-head attention mechanism. The experiment aimed to assess the attention mechanism's role in capturing key information from user reviews and course data, focusing on feature interaction and aggregation.

- **Ours-Attention**: In this variant, the multi-head attention mechanism was removed, retaining only the BiLSTM layer. The experiment evaluated the BiLSTM layer's ability to capture sequence dependencies and contextual information, particularly in semantic understanding and feature representation.

**Table 5. Ablation experiment results.**

| Model | RMSE | | | MAE | | |
|---|---|---|---|---|---|---|
| | Original | Real | Difference | Original | Real | Difference |
| **Ours** | 0.422 | 0.291 | 0.13 | 0.346 | 0.283 | 0.06 |
| **Ours-BiLSTM** | 0.549 | 0.478 | 0.07 | 0.419 | 0.316 | 0.10 |
| **Ours-Attention** | 0.522 | 0.328 | 0.19 | 0.355 | 0.304 | 0.05 |

Based on the experimental results presented in Table 5 and Fig 7, it can be observed that after the removal of both the BiLSTM layer and the attention mechanism, the RMSE and MAE values on both datasets increased, indicating that both components play a significant role in improving the model's performance. However, a comparative analysis reveals that the removal of the BiLSTM layer led to a more substantial decline in performance, particularly with a greater increase in RMSE and MAE on the original dataset. This suggests that the BiLSTM layer contributes more to the model's ability to capture sequence information and semantic dependencies. Therefore, while the attention mechanism is critical for enhancing feature interaction and recommendation accuracy, the BiLSTM layer plays a more crucial role in processing sequence information and semantic understanding.

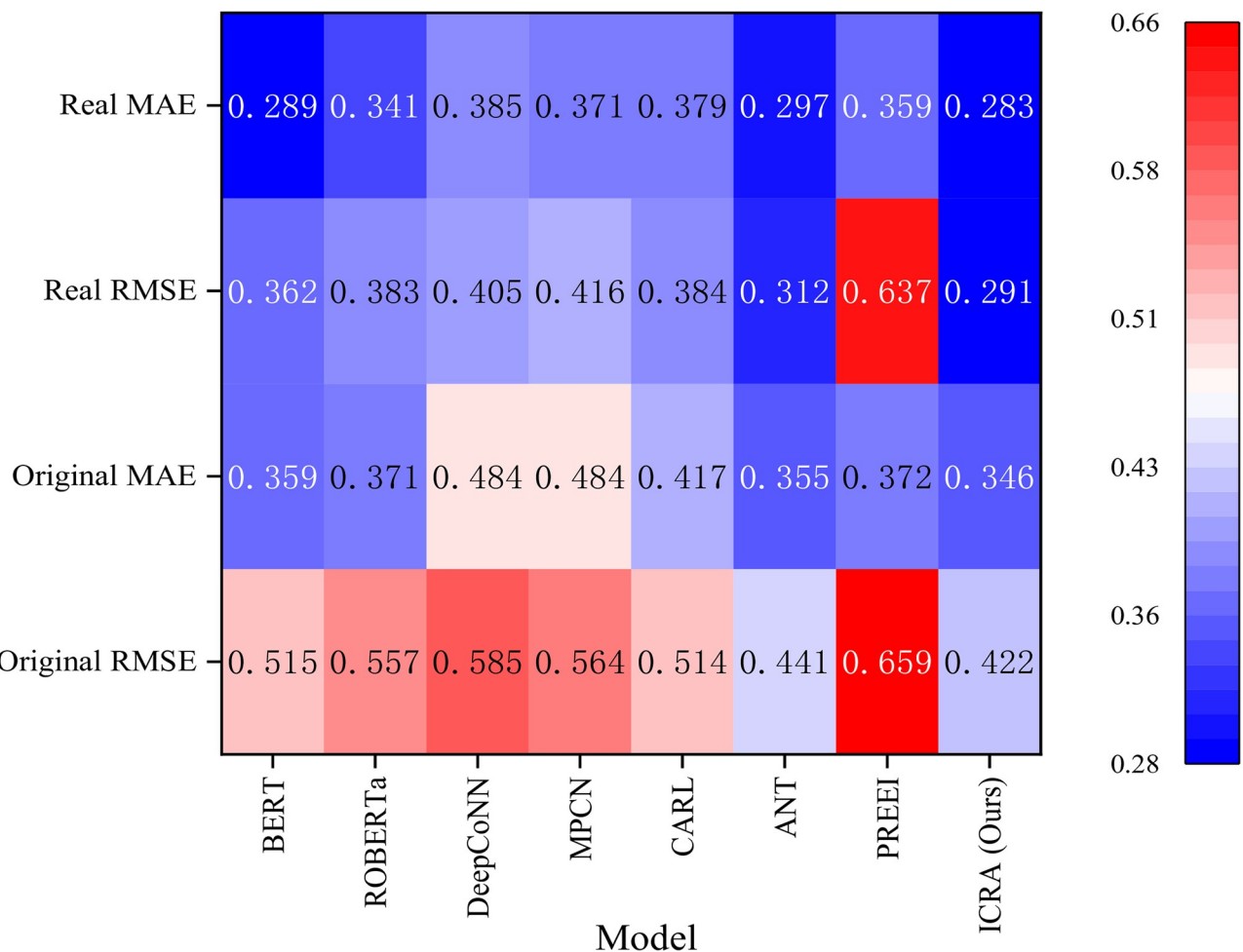

**Fig 7. Impact of components on the model.**

## Parameter sensitivity experiment

This section examines how varying the number of heads in multi-head attention and the dropout rate affect the performance of the ICRA algorithm. The experiment tests different configurations: the number of heads in multi-head attention {0, 2, 4, 6, 8} and dropout rates {0.1, 0.3, 0.5, 0.7, 0.9}. The performance results are illustrated in Fig 8.

The experiment results reveal a significant impact of attention heads on model performance. Without any attention mechanism (0 attention heads), the model shows higher RMSE and MAE, indicating poorer performance. Increasing the number of attention heads from 2 to 4 significantly enhances model predictive performance, resulting in notable reductions in RMSE and MAE. However, further increments to 6 and 8 attention heads result in diminished

(a)

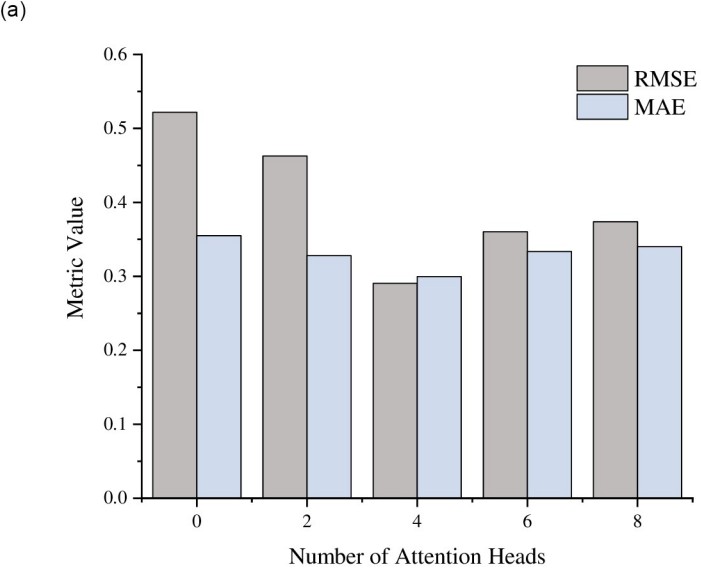

(b)

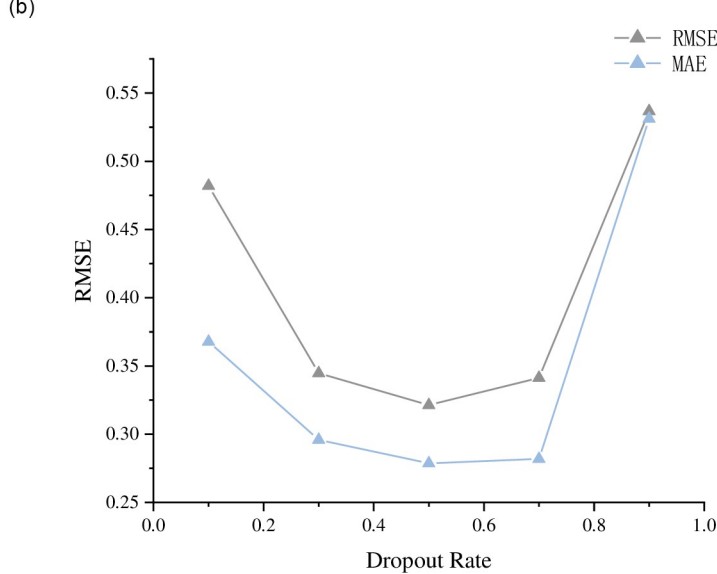

**Fig 8. The effect of model parameters on the model.** (a): Impact of multi-head attention heads. (b): Impact of Dropout.

performance, characterized by higher RMSE and MAE. This suggests that while a moderate number of attention heads improves model efficiency and accuracy, excessive attention heads may complicate the model, adversely affecting its performance.

Regarding the impact of dropout rates, the data shows that initially increasing the dropout rate improves and then diminishes model performance. Performance is relatively lower at a dropout rate of 0.1; significant improvements are observed at dropout rates of 0.3 and 0.5, reaching peak performance at 0.5. However, further increasing dropout rates to 0.7 and 0.9 results in significant performance declines, suggesting that high dropout rates may overly regularize the model, thus weakening its predictive ability. This study underscores the importance of considering suitable dropout rates and attention head numbers in model design to achieve a balance between model complexity and predictive performance. The research provides empirical support for optimizing course recommendation systems and offers valuable guidance for algorithm design in related applications.

## Conclusion

This study proposes and validates the ICRA model, which significantly improves the accuracy of course rating predictions and recommendations by integrating user reviews and course descriptions. The ICRA model not only captures the nuanced semantics in user reviews but also addresses information gaps by incorporating professional course descriptions, reducing the interference of emotional language. For new or less-reviewed courses, the course overview serves as a key information source, effectively mitigating the cold-start problem. By combining sentiment analysis with objective information, the model provides users with more interpretative recommendations, enhancing user satisfaction and trust.

Despite the strong performance of the ICRA model on MOOC platforms, it has some limitations. It was developed based on data from the Chinese University MOOC platform, and its generalization to other online learning platforms remains unverified. Additionally, the false review detection module relies on existing sentiment lexicons and the ERNIE 3.0 model, which may have limited accuracy in identifying inauthentic reviews. Furthermore, the complexity of the model and its high computational demands may limit its application in resource-constrained environments.

Future research should focus on enhancing the generalization capabilities of the ICRA model and improving the false review detection module by incorporating more external data and advanced algorithms. Additionally, introducing graph neural network technology to analyze interaction graphs between students and courses could further optimize the recommendation system. It is also recommended to adopt standard evaluation metrics for recommender systems to comprehensively assess the model's performance in different scenarios. In conclusion, the ICRA model offers an innovative and effective course recommendation solution for MOOC platforms, with great potential for further development, significantly improving user learning experiences and recommendation outcomes.

## Author Contributions

**Data curation:** Biao Yang.

**Formal analysis:** Yuqi Hou.

**Funding acquisition:** Bing Li.

**Investigation:** Jiangtao Dong.

**Methodology:** Bing Li.

**Validation:** Jiangtao Dong.

**Visualization:** Biao Yang.

**Writing – original draft:** Yuqi Hou.

**Writing – review & editing:** Xile Wang.

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
