## [Decision Letter · Decision Letter 0]

19 Sep 2024

PONE-D-24-27196ICRA: A Study of Highly Accurate Course Recommendation Models Incorporating False Review Filtering and ERNIE 3.0PLOS ONE

Dear Dr. Li,

Thank you for submitting your manuscript to PLOS ONE. After careful consideration, we feel that it has merit but does not fully meet PLOS ONE’s publication criteria as it currently stands. Therefore, we invite you to submit a revised version of the manuscript that addresses the points raised during the review process.

**ACADEMIC EDITOR: See comments below from the reviewers.**==============================

We look forward to receiving your revised manuscript.

Kind regards,

Muhammad Usman Tariq, Ph.D

PFHEA, CFCIPD, CMBE

SFSEDA, SMIEEE

Academic Editor

PLOS ONE

Journal requirements: 1. When submitting your revision, we need you to address these additional requirements. Please ensure that your manuscript meets PLOS ONE's style requirements, including those for file naming. The PLOS ONE style templates can be found at https://journals.plos.org/plosone/s/file?id=wjVg/PLOSOne_formatting_sample_main_body.pdf and https://journals.plos.org/plosone/s/file?id=ba62/PLOSOne_formatting_sample_title_authors_affiliations.pdf. 2. Please note that PLOS ONE has specific guidelines on code sharing for submissions in which author-generated code underpins the findings in the manuscript. In these cases, we expect all author-generated code to be made available without restrictions upon publication of the work. Please review our guidelines at https://journals.plos.org/plosone/s/materials-and-software-sharing#loc-sharing-code and ensure that your code is shared in a way that follows best practice and facilitates reproducibility and reuse 3. In your Methods section, please include additional information about your dataset and ensure that you have included a statement specifying whether the collection and analysis method complied with the terms and conditions for the source of the data.  4. We note that the grant information you provided in the ‘Funding Information’ and ‘Financial Disclosure’ sections do not match.  When you resubmit, please ensure that you provide the correct grant numbers for the awards you received for your study in the ‘Funding Information’ section. 5. Thank you for stating the following financial disclosure:  [The authors acknowledge the financial support provided by the National Natural 446Science Foundation of China (72161020), the Jiangxi Provincial Natural Science 447Foundation (20224BAB202023), the Jiangxi Social Science Foundation Project 448(21GL44), the Science and Technology Research Project of Jiangxi Provincial Education 449Department (GJJ2200333).].  Please state what role the funders took in the study.  If the funders had no role, please state: ""The funders had no role in study design, data collection and analysis, decision to publish, or preparation of the manuscript."" If this statement is not correct you must amend it as needed. Please include this amended Role of Funder statement in your cover letter; we will change the online submission form on your behalf. 6. Thank you for stating the following in the Acknowledgments Section of your manuscript: [The authors acknowledge the financial support provided by the National Natural 446Science Foundation of China (72161020), the Jiangxi Provincial Natural Science 447Foundation (20224BAB202023), the Jiangxi Social Science Foundation Project 448(21GL44), the Science and Technology Research Project of Jiangxi Provincial Education 449Department (GJJ2200333). This funding facilitated the completion of this research, for 450which we are sincerely grateful.]We note that you have provided funding information that is not currently declared in your Funding Statement. However, funding information should not appear in the Acknowledgments section or other areas of your manuscript. We will only publish funding information present in the Funding Statement section of the online submission form. Please remove any funding-related text from the manuscript and let us know how you would like to update your Funding Statement. Currently, your Funding Statement reads as follows:  [The authors acknowledge the financial support provided by the National Natural 446Science Foundation of China (72161020), the Jiangxi Provincial Natural Science 447Foundation (20224BAB202023), the Jiangxi Social Science Foundation Project 448(21GL44), the Science and Technology Research Project of Jiangxi Provincial Education 449Department (GJJ2200333).] Please include your amended statements within your cover letter; we will change the online submission form on your behalf. 7. When completing the data availability statement of the submission form, you indicated that you will make your data available on acceptance. We strongly recommend all authors decide on a data sharing plan before acceptance, as the process can be lengthy and hold up publication timelines. Please note that, though access restrictions are acceptable now, your entire data will need to be made freely accessible if your manuscript is accepted for publication. This policy applies to all data except where public deposition would breach compliance with the protocol approved by your research ethics board. If you are unable to adhere to our open data policy, please kindly revise your statement to explain your reasoning and we will seek the editor's input on an exemption. Please be assured that, once you have provided your new statement, the assessment of your exemption will not hold up the peer review process.

Reviewers' comments:

Reviewer's Responses to Questions

**Comments to the Author**

1. Is the manuscript technically sound, and do the data support the conclusions?

Reviewer #1: Yes

Reviewer #2: Partly

2. Has the statistical analysis been performed appropriately and rigorously? 

Reviewer #1: N/A

Reviewer #2: Yes

3. Have the authors made all data underlying the findings in their manuscript fully available?

Reviewer #1: Yes

Reviewer #2: Yes

4. Is the manuscript presented in an intelligible fashion and written in standard English?

Reviewer #1: Yes

Reviewer #2: Yes

5. Review Comments to the Author

Reviewer #1: 1. Please provide the full form of the acronym MOOC when it is first mentioned in the text.

2. It is recommended to place Table 3 after its corresponding explanation, similarly for Figure 2.

3. In Table 3, please clarify the meaning of the "Improved" column.

4. The legend in Figure 6 uses the terms "Actual" and "Original Data," which differ from the term "Real data" used throughout the paper. Please standardize the terminology for consistency.

Reviewer #2: The authors present a novel online course recommendation model that incorporates user reviews and course profiles for better recommendation. In addition, authors propose a fake comment detection model that identifies and filters reviews before inputting them into the recommendation model.

Overview:

The paper is easy to read and follow. The motivation of using the user reviews/comments in the recommendation model is presented clearly. The performance evaluation is thorough and shows performance improvement over state of the art. While the evaluation uses a single dataset, details on data creation and availability are reasonable and explained clearly.

However, the methods and experiments lack some salient details and needs several improvements (described below):

Major comments:

1) It’s not clear if the authors evaluate their method in which of the following settings:

a. Given a new user, find the most appropriate course recommendation? - If this is the case, the recommendation should be conditioned on some kind of user query/need. In addition. The model should be evaluated using standard recommendation model metrics such as precision@k, recall@K, Hit rate, Mean reciprocal rank (MRR) etc.

Vs

b. Given a course profile and its reviews , determine what is the rating the course should get? - if this is the objective, the proposed model should be presented as a course rating model and not a course recommendation model. (which I believe is the case)

2) The presence and filtering of ‘false reviews’: Authors do not present any statistics/ citations to support their argument about wide spread prevalence of fake reviews in online courses. More importantly, there is no evaluation provided to determine if the ‘false comments recognition’ module does in fact only remove fake reviews or just filters random reviews. While an argument can be made that the review filtering does improve performance, if it is in fact removing false reviews seems a claim that should be substantiated with better analysis.

3) Evaluation: The authors present their evaluation results and list “Improvement percentage” in the last line of Table 3 .The improvement percentage however is calculated by calculating “Average scores from all other benchmarks”. This is a gross misrepresentation of improvement data. Improvement is always calculated from the next best performing method and not an average of the chose benchmarks. Authors are strongly advised to update the manuscript to reflect the improvement from the best performing baseline method, which seem to be in the range of 2-5%.

4) The paper lacks following details in methods and experimental setup, which makes it hard to follow the results and conclusion:

a. The experimental setup does not provide any details on what is y_true and what is y_pred in context of their evaluation.

b. The ablation designs are not explained at all. They are just listed as “Ours-biLSTM” and “Ours- Attention” with no explanation of what layers of the model were dropped or changed to design ablation study.

c. Figure 5 Model architecture: Its not clear what is in the user embedding in Figure 5 and how is it constructed using filtered reviews. Did the authors mean “review/comment embedding” instead of “user embedding”

Minor comments :

• Figure 1 is seems like a collection of symbols without any textbox describing anything inside the figure or inside the actual text. Authors shoud remove Figure 1 or add appropriate textboxes in the figure and a description in figure caption to make it useful for the reader.

• Figure 3 , please list clearly what is the input to the model and what does the output tensor represent.

• Introduction (page 3/18), the line “Some studies have shown innovative approaches and results” is repeated.

6. PLOS authors have the option to publish the peer review history of their article (what does this mean?). If published, this will include your full peer review and any attached files.

Reviewer #1: **Yes: **ASRAFUL SYIFAA' AHMAD

Reviewer #2: No

---

## [Author Response · Author response to Decision Letter 0]

28 Oct 2024

Original Manuscript ID: PONE-D-24-27196 

Original Article Title: “ICRA: A Study of Highly Accurate Course Recommendation Models Incorporating False Review Filtering and ERNIE 3.0”

Dear Editors and Reviewers,

Thank you for giving us this valuable opportunity to submit a revised version of our manuscript titled "[ICRA: A Study of Highly Accurate Course Recommendation Models Incorporating False Review Filtering and ERNIE 3.0]" (Manuscript ID: PONE-D-24-27196). We sincerely appreciate the detailed and constructive comments provided by the reviewers. We have carefully revised the manuscript by incorporating all the reviewers' suggestions. Compared to the original manuscript, we have added some figures and explanations, and have thoroughly checked and corrected spelling and grammatical errors in the manuscript. We hope that the revisions in the manuscript, as well as our point-by-point responses to the reviewers' comments, will make our manuscript suitable for publication in PLOS ONE.

Attached to this letter is our point-by-point response to the reviewers' comments. For ease of discussion, we first restate your comments in blue, followed by our responses. Additionally, the changes in the manuscript are highlighted in red.

We look forward to hearing from you at your earliest convenience.

Yours sincerely,

Bing Li et al.

Reponses to Editor

Thank you for your valuable feedback and suggestions on our submitted manuscript. Based on your comments, we have made the following revisions to the manuscript:

1.Formatting Requirements：We noted that the current manuscript format does not meet the accepted file types for PLOS ONE. In response, we have reformatted the manuscript as a .tex file (with an accompanying .pdf) and have resubmitted it. Please review the updated files at your convenience.

2.Code Sharing Guidelines：We have checked and ensured that all codes related to the research results are publicly accessible and comply with PLOS ONE's code sharing guidelines. To ensure the reproducibility and reusability of the research, we are providing the access link to the code: https://github.com/hyq9/ICRA

3.Dataset Information and Statement：Thank you for your valuable feedback on my manuscript. In response to your suggestions, we have added detailed information about the dataset in the Methods section of the manuscript(Data collection, page 6), along with a statement confirming that our data collection and analysis processes strictly adhere to the terms and conditions of the data source to ensure legal and regulatory compliance.

4.Correction of Funding Information：We have carefully reviewed the "Funding Information" and "Financial Disclosure" sections to ensure the correct grant numbers are listed. All funding-related content has been removed from the acknowledgments section of the manuscript.

5.Funder Role Statement：The funder role statement has been updated. We have included the following new statement in the cover letter: "The funders had no role in study design, data collection and analysis, decision to publish, or preparation of the manuscript." This statement has also been reflected in the revised manuscript.

6.Removal of Funding Information from the Manuscript：Funding-related content has been removed from the acknowledgments section of the manuscript. Complete funding information is now provided in the "Funding Statement" section of the online submission form, which has been updated according to your request.

7.Data Availability Statement：Thank you for your recommendation regarding the data repository. To comply with PLOS’s data storage standards, we have uploaded the research dataset to the recommended public repository Figshare. The dataset’s access link and DOI are as follows: https://doi.org/10.6084/m9.figshare.27301371.v1

8.Reference List：We have thoroughly reviewed and updated the reference list to ensure its completeness and accuracy. For retracted articles, we have either provided the reason for citation in the main text or replaced them with current relevant literature. Additionally, if a retracted article was cited, we have clearly marked its retracted status in the reference list.

Reponses to Reviewer #1

Comment 1: 

Please provide the full form of the acronym MOOC when it is first mentioned in the text.

Response 1: 

Thank you very much for your reminder. We have already provided the full name of MOOC, "Massive Open Online Courses," when it is first mentioned to ensure that readers can clearly understand the term.

Revised 1 (Introduction, Page 1):

Concurrently, the proliferation of MOOC(Massive Open Online Courses) platforms like Coursera, edX, and Udacity has presented learners with numerous course options, posing both opportunities and challenges.

Comment 2: 

It is recommended to place Table 3 after its corresponding explanation, similarly for Figure 2.

Response 2:

Thank you for your valuable suggestions. We have adjusted the placement of Table 4 (formerly Table 3) and Figure 2 based on your comments, positioning them after the corresponding explanations. In addition, we have refined the surrounding text to ensure coherence and logical flow, making it easier for readers to follow the discussion. Please refer to Revised 2 for the adjustments made to Figure 2 and Revised 3 for the modifications to Table 4. 

Revised 2 (Introduction, Page 3):

Fig 2 compares traditional recommender systems with the ICRA recommendation model. Traditional recommender systems primarily rely on user reviews and ratings, but inconsistencies between reviews and ratings often affect system reliability. To address this, the ICRA model integrates false review filtering with course overview analysis, improving recommendation accuracy and reliability.

Comment 3: 

In Table 3, please clarify the meaning of the "Improved" column.

Response 3:

Thank you for your thorough review of our paper. In response to the question regarding the "Improved" column in Table 4 (formerly Table 3), we have made the following adjustments in the revised manuscript: The original table used the "Improved" column to display the performance improvements of each model in terms of RMSE and MAE on raw and real data, but the numerical presentation may not have been sufficiently intuitive. In the revision, we replaced the "Improved" column with a "Difference" column, which directly shows the differences between the raw data and real data for both RMSE and MAE. Additionally, we have included a heatmap that visually represents the performance changes of different models using color gradients. These adjustments help to more clearly illustrate the performance changes of each model before and after filtering fake reviews, making it easier for readers to intuitively understand the extent of model improvements. 

Revised 3 (Comparison experiment，Page 13):

Table 4 compares the RMSE and MAE of each model on the original and real datasets, demonstrating performance improvements. The "Difference" column represents the discrepancy between the original and real datasets, reflecting the performance changes after filtering out false reviews. The last row, "Improved" shows the percentage improvement of the IORA model compared to the historically best methods. On the original dataset, IORA's RMSE improved by 4.31% over ANT, while MAE improved by 2.54%. On the real dataset, RMSE improved by 6.73% over ANT and MAE improved by 2.08% over BERT. The heatmap in Fig 6 provides a visual representation of model performance, where darker colors indicate better performance. Most models showed improvements on the real dataset, with the IORA model showing the most significant RMSE and MAE improvements, as highlighted by the dark blue regions. These results demonstrate that the IORA model achieved significant progress in predictive accuracy, particularly in addressing false reviews.

Compared to ICRA, the improvements in other models are relatively modest. DeepCoNN and MPCN show notable performance gains on real data after filtering out false comments, whereas PREEI exhibits minimal improvement, suggesting limitations in adaptability. The strong performance of the ANT model underscores the effectiveness of its attention mechanism in managing complex recommendation tasks.

The ICRA model’s improvements in RMSE and MAE surpass those of most other models, demonstrating its robustness and accuracy in course recommendation tasks. This indicates that ICRA effectively captures deep user-course interactions, delivering precise recommendations even in challenging, noisy datasets.

Comment 4: 

The legend in Figure 6 uses the terms "Actual" and "Original Data," which differ from the term "Real data" used throughout the paper. Please standardize the terminology for consistency.

Response 4:

Thank you for pointing out the inconsistency. We have made adjustments in the revised manuscript to the legend of Figure 7 (formerly Figure 6) and the corresponding content in the text, consistently using the term "Real data" throughout to ensure uniformity and avoid reader confusion.

Revised 4 (Ablation experiment, Page 15):

Reponses to Reviewer #2

Major comments 1:

It’s not clear if the authors evaluate their method in which of the following settings:

a. Given a new user, find the most appropriate course recommendation? - If this is the case, the recommendation should be conditioned on some kind of user query/need. In addition. The model should be evaluated using standard recommendation model metrics such as precision@k, recall@K, Hit rate, Mean reciprocal rank (MRR) etc.

Vs

b. Given a course profile and its reviews , determine what is the rating the course should get? - if this is the objective, the proposed model should be presented as a course rating model and not a course recommendation model. (which I believe is the case)

Response 1: 

Thank you very much for your thorough review of our manuscript, especially for the specific suggestions regarding the evaluation settings. Your feedback has been extremely valuable in improving the content of our paper. In response to the model positioning issue you raised, we have made the following clarifications and revisions:

Firstly, the primary goal of our model is to predict the rating a course should receive based on course materials and user reviews. From this perspective, our model aligns more with the definition of a course rating prediction model rather than a traditional Top-K recommendation system. By performing deep semantic analysis of course reviews, we can accurately predict user ratings for the courses. Therefore, in the model evaluation, we used RMSE and MAE as the main metrics to measure the accuracy of rating predictions, rather than the evaluation metrics commonly used in Top-K recommendation systems (such as precision@k, recall@k, etc.).

Additionally, recommendation systems can be categorized based on different task scenarios, such as rating prediction, Top-K recommendation, click-through rate prediction, context-aware recommendation, and group recommendation. Each task corresponds to different evaluation metrics: rating prediction typically uses RMSE and MAE, Top-K recommendation uses Precision@K, Recall@K, NDCG, and click-through rate prediction commonly uses AUC and CTR. The nature of the task determines the choice of evaluation metrics, which helps comprehensively assess the performance of the recommendation system. In rating prediction tasks, RMSE and MAE effectively reflect the error between predicted and actual ratings. Lower RMSE and MAE values indicate smaller prediction errors, thus indicating better model performance.

However, as you mentioned, rating predictions can also be further used to recommend suitable courses to users. In the ICRA model, rating prediction is one of the key steps to improving recommendation accuracy, rather than the final goal. Therefore, the model not only predicts ratings but can also recommend the best courses to users by ranking or filtering based on the predicted ratings. By predicting ratings for different courses, the model helps users make more informed decisions and thus has a certain recommendation system function.

Although we have not yet used standard recommendation system evaluation metrics, we plan to further expand the model’s functionality in the future to generate more personalized course recommendations by incorporating users' historical behaviors and preferences. In the revised manuscript, we have adjusted the model architecture and conclusion sections to more clearly demonstrate the model's dual functionality: not only as a rating model but also as a tool to provide personalized course recommendations to users.

Once again, thank you for your valuable feedback. Your insights have been crucial in helping us improve the content of the paper. We look forward to presenting the full functionality of the model and addressing the issues you raised in the revised manuscript.

Revised 1:

(Model architecture, Page 9)

Rating prediction is one of the core problems in personalized recommendation systems, aiming to score the interactions between users and courses. The ICRA model predicts user ratings for unseen courses, providing a basis for personalized ranking in the recommendation system. The model predicts scores for all candidate courses and ranks them based on the predicted scores, prioritizing those with higher ratings to achieve personalized recommendations. The ICRA model leverages data collected from the Chinese University MOOC platform, combined with a self-built sentiment lexicon and the pre-trained ERNIE 3.0 model, to conduct in-depth semantic analysis. This model not only accurately identifies and filters fake reviews, ensuring high-quality datasets, but also predicts user ratings for new courses based on their historical rating records. To achieve this, the ICRA model consists of an embedding layer, a feature processing layer, a regularization layer, and an output layer, all of which work closely together to improve rating prediction accuracy.

Figure 5 illustrates the overall architecture of the ICRA model, where represents the user comment document, represents the course overview document, and are the embedding matrices for user and course comment documents, respectively, and and denote the actual and predicted ratings of users for courses.

(Conclusion, Page 16)

Despite the strong performance of the ICRA model on MOOC platforms, it has some limitations. It was developed based on data from the Chinese University MOOC platform, and its generalization to other online learning platforms remains unverified. Additionally, the false review detection module relies on existing sentiment lexicons and the ERNIE 3.0 model, which may have limited accuracy in identifying inauthentic reviews. Furthermore, the complexity of the model and its high computational demands may limit its application in resource-constrained environments.

Major comments 2:

The presence and filtering of ‘false reviews’: Authors do not present any statistics/ citations to support their argument about wide spread prevalence of fake reviews in online courses. More importantly, there is no evaluation provided to determine if the ‘false comments recognition’ module does in fact only remove fake reviews or just filters random reviews. While an argument can be made that the review filtering does improve performance, if it is in fact removing false reviews seems a claim that should be substantiated with better analysis.

Response 2: 

Thank you for your detailed feedback and valuable suggestions on our paper. We recognize the lack of sufficient statistical data and citations to support our argument regarding the prevalence of fake reviews. Furthermore, the analysis of the effectiveness of the fake review detection module is indeed insufficient, as it fails to demonstrate whether the module only removed fake reviews rather than random reviews. In response to your comments, we plan to make the following improvements in the revised manuscript:

1.Citing Relevant Research and Statistical Data: In the revised manuscript, we will cite relevant research and reports, providing statistical data to support the prevalence of fake reviews on online education platforms (such as MOOC platforms). The data will come from published surveys and market an

---

## [Editor Report · Decision Letter 1]

4 Nov 2024

ICRA: A Study of Highly Accurate Course Recommendation Models Incorporating False Review Filtering and ERNIE 3.0

PONE-D-24-27196R1

Dear Dr. Li,

We’re pleased to inform you that your manuscript has been judged scientifically suitable for publication and will be formally accepted for publication once it meets all outstanding technical requirements.

Kind regards,

Muhammad Usman Tariq, Ph.D

PFHEA, CFCIPD, CMBE

SFSEDA, SMIEEE

Academic Editor

PLOS ONE

---

## [Editor Report · Acceptance letter]

29 Nov 2024

PONE-D-24-27196R1 

PLOS ONE

Dear Dr. Li, 

I'm pleased to inform you that your manuscript has been deemed suitable for publication in PLOS ONE. Congratulations! Your manuscript is now being handed over to our production team.

Kind regards, 

on behalf of

Dr. Muhammad Usman Tariq 

Academic Editor

PLOS ONE